# Adaptive Neural Ranking Framework: Toward Maximized Business Goal for Cascade Ranking Systems

## ABSTRACT

Cascade ranking is widely used for large-scale top-k selection problems in online advertising and recommendation systems, and learning-to-rank is an important way to optimize the models in cascade ranking systems. Previous works on learning-to-rank usually focus on letting the model learn the complete order or pay more attention to the order of top materials, and adopt the corresponding rank metrics (e.g. NDCG@k and OAP) as optimization targets. However, these optimization targets can not adapt to various cascade ranking scenarios with varying data complexities and model capabilities; and the existing metric-driven methods such as the Lambda framework can only optimize a rough upper bound of the metric, potentially resulting in performance misalignment. To address these issues, we first propose a novel perspective on optimizing cascade ranking systems by highlighting the adaptability of optimization targets to data complexities and model capabilities. Concretely, we employ multi-task learning framework to adaptively combine the optimization of relaxed and full targets, which refers to metrics Recall@m@k and OAP respectively. Then we introduce a permutation matrix to represent the rank metrics and employ differentiable sorting techniques to obtain a relaxed permutation matrix with controllable approximate error bound. This enables us to optimize both the relaxed and full targets directly and more appropriately using the proposed surrogate losses within the deep learning framework. We named this method as Adaptive Neural Ranking Framework (abbreviated as ARF). Furthermore, we give a specific practice under ARF. We use the NeuralSort method to obtain the relaxed permutation matrix and draw on the uncertainty weight method in multi-task learning to optimize the proposed losses jointly. Experiments on a total of 4 public and industrial benchmarks show the effectiveness and generalization of our method, and online experiment shows that our method has significant application value.

## CCS CONCEPTS

• **Do Not Use This Code → Generate the Correct Terms for Your Paper**; *Generate the Correct Terms for Your Paper*; Generate the Correct Terms for Your Paper; Generate the Correct Terms for Your Paper.

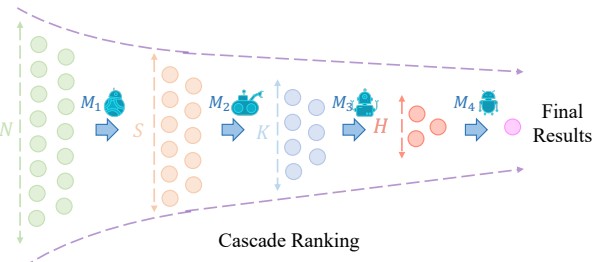

**Figure 1: A classic cascade ranking architecture, which includes four stages: Matching, Pre-ranking, Ranking, and Re-ranking.**

## KEYWORDS

Learning to Rank in Cascade Systems, Differentiable Sorting, Multi-task Learning

**ACM Reference Format:**

Anonymous Author(s). 2018. Adaptive Neural Ranking Framework: Toward Maximized Business Goal for Cascade Ranking Systems. In *Woodstock '18: ACM Symposium on Neural Gaze Detection, June 03–05, 2018, Woodstock, NY*. ACM, New York, NY, USA, 12 pages. https://doi.org/10.1145/1122445.1122456

## 1 INTRODUCTION

The cascade ranking [31, 40, 48] has garnered increasing research interest and attention as a mature solution to large-scale ranking and top-k set selection problems under limited resources. It is widely used in business systems, such as online advertising and recommendation systems, which have an important impact on human production and life. Take typical online advertising systems as an example, they often employ a cascade ranking architecture with four stages, as illustrated in Figure 1. When an online request arrives, the Matching stage first selects a subset of ads from the entire ad inventory (with a magnitude of $N$), typically of size $S$. Subsequently, the Pre-ranking stage predicts the value of these $S$ ads and selects the top $K$ ads to send to the Ranking stage. This process continues iteratively until the last stage of the cascade ranking system decides the ads for exposure.

In cascade ranking systems, letting one stage learn from its post-stages through learning-to-rank (LTR) is an important method to maximize system traffic efficiency, and it is also one of the most commonly used methods in the industry. Take the Pre-ranking stage as an example, people usually randomly sample some ads in the Pre-ranking space and adopt learning-to-rank methods to let the Pre-ranking model learn the order of these ads produced by the Ranking model. Traditional learning-to-rank methods (whether the typical point-wise methods[11, 12, 26, 35], pair-wise methods[4, 7, 15, 46, 56], or list-wise methods[8, 39, 47, 48, 53]) often focus on the entire order or top-k order of the training data, which refer to the ranking metrics such as $NDCG$ and $NDCG@k$. When we directly adopt these methods, we actually try to drive the system

towards the oracle condition, which is the most strict sufficient, and unnecessary condition to maximize the business goal of the cascade ranking system. In short, the oracle condition (formulated in section 4.1.) requires each stage to have an idealized model that can give the ground-truth output regardless of the input.

A fact in cascade ranking systems is that model complexity typically increases from the front to the end stages, and so does the models' capacity. This means that it may be impossible for a stage to completely fit its post-stage, even if the training data is down-sampled from its post-stage. In real cascade ranking systems, there is often a large gap between the prediction capabilities of a certain stage and its post-stage. When we adopt traditional learning-to-rank methods to optimize towards the "Oracle Condition", we cannot guide the model to only lose the sufficiency but not the necessity of the actually achieved condition when the model makes mistakes. Therefore, when facing the situation that the training data is too complex for the model, it's better to optimize the model towards some relaxed conditions. To this end, we propose the "stage recall complete condition" as the relaxed condition, which only requires the $Recall@m@k$ of each stage to be equal to 1 (cf., Section 4.1).

To optimize the model towards the relaxed condition, an intuitive thought is to relax the existing learning-to-rank methods. Wang et al.(2018)[43] extend LambdaRank[3] to a unified framework for designing metric-driven surrogate loss and propose LambdaLoss for optimizing $NDCG$ under this framework. Although we can give the surrogate loss for $Recall@m@k$ following [43], such a surrogate loss can only optimize a rough upper bound of the metric. In order to better optimize $Recall@m@k$, we first introduce an alternative perspective to describe sorting based on permutation matrix and matrix operation, and re-formulate $Recall@m@k$. Furthermore, we introduce differentiable sorting techniques that can produce a relaxed permutation matrix with a controllable approximate error bound of the hard permutation matrix. Concretely, we adopt NeuralSort[18] to produce an unimodal row stochastic permutation matrix, and design the novel surrogate loss $L_{Relax}$ for optimizing $Recall@m@k$ directly. With the temperature $\tau$ of NeuralSort, we can control the relaxation of the permutation matrix.

In cascade ranking systems, sometimes we may face another opposite situation that the data is simple enough for the model's capabilities, since the drawn samples of the post-stages may be in a small amount. At this time, just optimizing $L_{Relax}$ may not be good enough, and harnessing the relaxed target with the information of all pairs is more likely to be beneficial. To mine the information of all pairs, we also define a loss function $L_{Global}$ based on the relaxed permutation matrix which requires complete order accuracy and corresponds to the metric $OPA$ (cf., Eq 2). However, it is difficult to determine which situation belongs to the actual system before applying a specific loss function, and the optimal direction of the optimization should lie somewhere in between these two extremes. In order to build a robust method for various scenarios of cascade ranking systems, we employ the multi-task learning framework to empower the model to adaptively learn from $L_{Relax}$ and $L_{Global}$. We hope that $L_{Global}$ can be used as an auxiliary loss function to always help the model learn from $L_{Relax}$ more effectively. We name this approach as Adaptive Neural Ranking Framework (abbreviated as ARF). As a practice of ARF, we employ a simple gradient-based

optimization strategy named uncertainty-weight[24] and make a variation of this method to highlight the primacy of $L_{Relax}$.

To verify the effectiveness and generalization of our methods, we conducted comprehensive experiments on four datasets. Two datasets are constructed from a real-world online cascade ranking system. The other two are publicly available benchmark datasets, which are standard LTR datasets. Experiments show that even the surrogate loss $L_{Relax}$ significantly outperforms the baseline methods on $Recall@m@k$ and the complete ARF method can bring further improvements. We also compare the results of $L_{Relax}$ and the baseline methods which can specify the optimization for Recall when optimizing different $m$ and $k$. The results show that $L_{Relax}$ achieves overall better results under various $m$ and $k$ and shows higher consistency with $Recall@m@k$. Furthermore, we deployed ARF in an online advertising system and achieved significant improvements in business metrics, demonstrating that our proposed approach has significant commercial value.

In general, our main contributions are three-fold: 1) We propose a novel surrogate loss $L_{Relax}$ for better optimization of $Recall@m@k$, which is considered to be a more relevant metric to the effect of the cascade ranking system. 2) Considering the complexity of the model and data combination in cascade ranking systems, we propose ARF that utilizes the multi-task learning framework to harness the $L_{Relax}$ with the full pairs information by auxiliary loss $L_{Gobal}$ for building a robust learning-to-rank paradigm. 3) We conduct comprehensive offline experiments to verify the effectiveness and generalization of our method. We also deployed our method on an online cascade ranking system to study the impact of ARF on real-world applications.

## 2 RELATED WORK

### 2.1 Learning to Rank and Cascade Ranking

Learning to rank (LTR) [6, 8, 26, 33, 39, 48, 50] is a subdomain of machine learning and information retrieval that focuses on developing algorithms and models to improve the ranking of items in a list based on their relevance to a specific query or context. Extensive related work in this area spans several decades and includes both traditional and modern approaches. Traditional methods, such as ranking SVMs and RankNet[5], have provided foundational insights into pairwise and pointwise ranking techniques. More recent advancements have seen the adoption of deep learning, with neural network-based architectures like RankNet[5] and LambdaRank[3]. Wang et al.(2018)[43] further extend LambdaRank to Lambda framework for metric-driven loss designing. Jagerman et al.(2022)[22] point out that the LambdaLoss under Lambda framework lacks in optimizing NDCG@k and further proposed LambdaLoss@k. Additionally, research has delved into incorporating diverse features, handling multi-modal data, and addressing challenges in large-scale, dynamic, and personalized ranking scenarios. LTR continues to evolve, driven by the ever-increasing demand for efficient and effective information retrieval systems in fields like e-commerce, search engines, and recommendation systems.

Cascade ranking is widely used in large-scale top-k selection systems such as RankFlow [31, 40, 41]. The concept of multi-stage cascade ranking is introduced to strike a balance between the efficiency and effectiveness of a ranking system. The previous works

[9, 16, 27] on cascade ranking systems primarily focus on the assignment of different rankers to each stage to collectively achieve the desired trade-off. Recently, RankFlow [31] and CoRR [21] propose to train rankers on their specific data distributions to exploit the interactions between each stage. Advanced learning-to-rank methods are often designed for NDCG (Normalized Discounted Cumulative Gain), which is a widely used evaluation metric in information retrieval and recommendation systems. NDCG [44] takes into account both the relevance and the position of items in a ranked list, providing a more nuanced and accurate assessment of ranking quality. In this work, we designed a novel learning-to-rank method considering the properties of cascade ranking to achieve robust Recall optimization in different data cascade ranking scenarios.

## 2.2 Differentiable Sorting

Recently, differentiable approximations of the sorting function were introduced by Grover et al.(2019)[18]. They propose NeuralSort, a continuous relaxation of the argsort operator, which relaxes hard permutation matrices by approximating them as unimodal row-stochastic matrices. This relaxation allows for gradient-based stochastic optimization. Grover et al.(2019)[18] first applied NeuralSort to classification tasks such as four-digit MNIST classification and yielded good results. Cuturi et al.(2019)[13], Blondel et al.(2020)[1] and Petersen et al.(2021)[29] have successively proposed better methods for the approximation of hard sort. Petersen et al.(2022)[30] further proposed monotonic differentiable sorting networks based on DiffSort[29]. Swezey et al.(2021)[37] proposed a method based on NeurlSort to handle large-scale top-k sorting. When these works are applied to sorting or classification tasks, they usually use a straight-forward cross-entropy loss to minimize the difference between the permutation matrices of the labels and estimated results. In this work, we adopt the differentiable sorting technique to obtain the relaxation of permutation matrices and propose a novel loss based on permutation matrices for optimizing the Recall metric.

## 2.3 Multi-task Learning

Multi-task learning (MTL) trains a model on multiple related tasks, promoting shared representation and enhancing performance generalization. It has been effectively applied in various machine learning applications, such as natural language processing [49, 51, 55], computer vision [23, 54] and recommendation systems[20, 36, 38]. In the context of ranking, multitasking [19, 42] refers to the concurrent evaluation of various criteria or attributes to ascertain the order or relevance of items, such as search results or product listings. Contemporary ranking algorithms often incorporate a diverse array of factors. These include user preferences, click-through rates, the quality of content, its recency, and relevance, all aimed at providing more accurate and personalized rankings. By employing multitasking techniques, these algorithms can assign appropriate weight to each criterion, adapt to evolving user behavior, and strike a balance between competing objectives such as diversity and precision. This strategy ensures that the ranked results are specifically tailored to meet the unique needs and preferences of users. Furthermore, it maintains a dynamic and adaptive ranking system that continually evolves to meet the changing demands of users and the availability of content. In this work, we decompose an optimization problem

into a joint optimization problem of two similar sub-objectives and hope to utilize the multi-task method to adaptively find the optimal gradient direction for the original optimization problem. Inspired by the gradient-based multi-task learning methods[10, 20, 24, 45, 52], we design a variant of the uncertainty weight method[24] that emphasizes the primacy of one certain objective.

## 3 PROBLEM FORMULATION

Let $\mathcal{M}_i$ denote the $i$-th stage and its model of the cascade ranking system, $Q_i$ denote the sample space of $\mathcal{M}_i$, $Q_i$ denote the size of $Q_i$. Let $\mathcal{I}$ denote the impression space of the system and its size is recorded as $Q_{\mathcal{I}}$. $\mathcal{I}$ can be seen as a virtual post-stage of the system's end-stage. The number of stages is denoted by $T$. For figure 1, the Matching stage is $\mathcal{M}_1$, the Re-ranking stage is $\mathcal{M}_4$, and the $T$ equals to 4. Let $\mathcal{F}_{\mathcal{M}}^{\downarrow}(\mathcal{S})$ denote the ordered terms vector sorted by the score of model $\mathcal{M}$ in descending order, and $\mathcal{F}_{\mathcal{M}}^{\downarrow}(\mathcal{S})[: K]$ denote the top $K$ terms of $\mathcal{F}_{\mathcal{M}}^{\downarrow}(\mathcal{S})$. We use **Ocl** denoted the oracle model, which can definitely make a correct prediction, even though such a model may not exist in reality.

The task of learning-to-rank methods are to optimize the models, so that the cascade ranking system can produce a better impression set, namely $\mathcal{F}_{\mathcal{M}_4}^{\downarrow}(\mathcal{F}_{\mathcal{M}_3}^{\downarrow}(\mathcal{F}_{\mathcal{M}_2}^{\downarrow}(\mathcal{F}_{\mathcal{M}_1}^{\downarrow}(Q_1)[: Q_2])[: Q_3])[: Q_4])[: Q_{\mathcal{I}}]$. Although the linkage influence of different stages is also an important factor affecting the final exposure quality in the cascade ranking system, we primarily focus on improving the learning of individual stages, rather than analyzing the impact of different stages on each other, which is beyond the scope of this work. In other words, when we optimize $\mathcal{M}_i$, all $\mathcal{M}_j$ for $j \neq i$ are regarded as static. Besides, LTR methods usually can be only used for $\mathcal{M}_{i<T}$, and assume that $\mathcal{M}_T$ or $\mathcal{M}_{i<j\leq T}$ is the **Ocl** or satisfying other optimal assumptions, let the model $\mathcal{M}_i$ learn the data produced by $\mathcal{M}_T$ or $\mathcal{M}_{i<j\leq T}$. Although these assumptions may be not to hold in reality, and learning from $\mathcal{M}_T$ or $\mathcal{M}_{i<j\leq T}$ will be affected by the bias of the sample selection problem and the model itself, our work mainly focuses on how to make LTR better fit the data produced by the system, so some debiased LTR methods are not discussed in this paper and are not within the comparison range.

In the following, we formulate the problem and explain our approach mainly based on the pre-reranking stage of the cascade ranking system shown in Figure 1; the extension to other cascade ranking systems and stages is straightforward. Now we formulate the common settings of learning-to-rank in cascade ranking. Let $D_{train}$ be the training set, which can be formulated as:

$$D_{train} = \{(f_{u_i}, \{f_{a_i^j}, v_i^j | 1 \leq j \leq n\})_i\}_{i=1}^N \tag{1}$$

where $u_i$ means the user of the $i$-th impression in the training set, $a^j$ means the $j$-th material for ranking in the system. $N$ is the number of impressions of $D_{train}$. The $i$ in $a_i^j$ means that the sample $a^j$ corresponds to impression $i$. The size of the materials for each impression is $n$. $f_{(\cdot)}$ means the feature of $(\cdot)$. $u_i$ and $a_i^j$ are drawn i.i.d from the space $Q_2$. $v_i^j$ is considered to be the ground truth value (the higher value is considered better) of the pair $(u_i, a^j)$, which can have many different specific forms. $v$ can be the rank index which is the relevance position (in descending order) in the system when the

request happens. It can also be uniformly scored by $\mathcal{M}_{t \geq 3}$ through some exploration mechanisms. For example, let $\mathcal{M}_T$ (namely $\mathcal{M}_4$ in Figure 1) uniformly scores the sampled samples. Let $\mathcal{M}_2(u_i, a_i^j)$ denote the score predict by $\mathcal{M}_2$ on the pair $(u_i, a_i^j)$ and $\mathcal{M}_2^{(i,j)}$ be the short for $\mathcal{M}_2(u_i, a_i^j)$. In section 4, we will discuss how to make $\mathcal{M}_2$ learn better from $D_{train}$.

## 4 APPROACH

In this section, we first discuss the oracle condition and the relaxation of the oracle condition for cascade ranking systems. Then we give a novel surrogate loss named $L_{Relax}$ to better optimize the model towards the relaxed condition. Finally, we describe the Adaptive Neural Ranking Framework, which aims to achieve robust learning-to-rank in various cascade ranking scenarios.

### 4.1 The Relaxation of Learning Targets for Cascade Ranking

With the notations in section 3, we can formulate the different assumed conditions for the system, such as the following most common condition, which is also the goal to be optimized when the cascade ranking systems use the LTR methods in the traditional way.

DEFINITION 1 (THE ORACLE CONDITION FOR CASCADE RANKING SYSTEMS). *When the cascade ranking system is met the oracle condition, it satisfies: 1) $\mathcal{M}_T$ is the **Ocl**, 2) for each $i<T$, $\mathcal{F}_{\mathcal{M}_i}^{\downarrow}(Q_1)$ equals to $\mathcal{F}_{\mathcal{M}_{i+1}}^{\downarrow}(Q_1)$.*

Obviously, the system met the oracle condition means each stage of the system has an oracle model. The goal of traditional learning-to-rank applications for cascade ranking can be viewed as optimizing the ordered pair accuracy (OPA) or NDCG to 1. The OPA and NDCG on $\mathcal{M}_2$ and the $i$-th impression of $D_{train}$ can be formulated as ($\mathcal{D}$ is the short for $D_{train}$):

$$OPA_{\mathcal{M}_2, \mathcal{D}[i]} = \frac{2 \sum_j^n \sum_{k=j+1}^n \mathbf{1}((\mathcal{M}_2^{(i,j)} - \mathcal{M}_2^{(i,k)})(v_i^j - v_i^k) \geq 0)}{n(n-1)} \tag{2}$$

$$
\begin{aligned}
NDCG_{\mathcal{M}_2, \mathcal{D}[i]} &= \sum_j^n \frac{1}{maxDCG_i} \frac{G_{i,j}}{D_{i,j}} \\
&= \sum_i^N \frac{1}{maxDCG_i} \sum_j^n \frac{2^{l_{i,j}} - 1}{log_2(p_{i,j}+1)} \\
l_{i,j} &= \pi(\mathcal{F}_{\mathbf{v}_i}^{\downarrow}(\mathcal{D}[i]), j) \\
p_{i,j} &= \pi(\mathcal{F}_{\mathcal{M}_2(u_i, \mathbf{a}_i^{(\cdot)})}^{\uparrow}(\mathcal{D}[i]), j) \\
maxDCG_i &= \sum_j^n \frac{2^{l_{i,j}} - 1}{log_2(p_{i,j}^* + 1)} \\
&= \sum_j^n \frac{2^{\pi(\mathcal{F}_{\mathbf{v}_i}^{\downarrow}(\mathcal{D}[i]), j)} - 1}{log_2(\pi(\mathcal{F}_{\mathbf{v}_i}^{\uparrow}(\mathcal{D}[i]), j) + 1)}
\end{aligned} \tag{3}
$$

where $\pi(\mathcal{F}_{\mathbf{v}_i}^{\downarrow}(\mathcal{D}[i]), j)$ means the rank index of the term $v_i^j$ in the vector $\mathcal{F}_{\mathbf{v}_i}^{\downarrow}(\mathcal{D}[i])$. Note that $\mathcal{F}^{\downarrow}$ and $\mathcal{F}^{\uparrow}$ denote sorting operators in descending order and ascending order respectively, and $\mathcal{F}_{\mathbf{v}_i}^{\downarrow}(\mathcal{D}[i])$ refers to the vector of $\mathcal{D}[i]$ sorted by the score vector $\mathbf{v}_i$. The same applies to $\pi(\mathcal{F}_{\mathcal{M}_2(u_i, \mathbf{a}_i^{(\cdot)})}^{\uparrow}(\mathcal{D}[i]), j)$.

Compared to the oracle condition, here we give a relaxed condition family named "stage recall complete condition". It has a scalable factor that can determine the degree of relaxation.

DEFINITION 2 (STAGE RECALL COMPLETE CONDITION). *When the cascade ranking system is met the condition, it satisfies 1) $\mathcal{F}_{\mathcal{M}_T}^{\downarrow}(Q_1)[: Q_I]$ equals to $\mathcal{F}_{\mathbf{Ocl}}^{\downarrow}(Q_1)[: Q_I]$, 2) for each $i<T$, $\mathcal{F}_{\mathcal{M}_{i+1}}^{\downarrow}(Q_1)[: m] \in \mathcal{F}_{\mathcal{M}_i}^{\downarrow}(Q_1)[: Q_{i+1}]$ for a certain $m$ that $m < Q_{i+1}$.*

When $Q_I \leq m \leq Q_{i+1}$, the "Stage Recall Complete Condition" is a sufficient condition for the cascade ranking system to achieve optimality. In particular, when $m = Q_i$, the "Stage Recall Complete Condition" is the necessary and sufficient condition for that. We can use $Recall@m@k$ in Eq 4 to characterize the extent to which this condition is achieved.

$$
\begin{aligned}
Recall_{\mathcal{M}_2, \mathcal{D}[i]}@m@k &= \frac{1}{k} \sum_j^n \mathbf{1}(a_i^j \in RS_i^m)\mathbf{1}(a_i^j \in GS_i^k) \\
RS_i^m &= \mathcal{F}_{\mathcal{M}_2^{(i,j)}}^{\downarrow}(\mathcal{D}[i])[: m] \qquad GS_i^k = \mathcal{F}_{\mathcal{M}_2^{(i,j)}}^{\downarrow}(\mathcal{D}[i])[: k]
\end{aligned} \tag{4}
$$

where $RS_i^m$ means the ordered recall set with size $m$ produced by $\mathcal{M}_2$ and $GS_i^k$ means the ground-truth set with size $k$ ordered by the score vector $\mathbf{v}_i$. $\mathbf{1}(\cdot)$ is the indicator function. Unlike traditional $Recall$, $Recall@m@k$ has a scaling factor $m$ that can specify the size of the ground-truth set and a scaling factor $k$ that can specify the size of the support set. When $Recall@m@k$ is optimized to 1 and satisfies $Q_{i+1} \leq k \leq m$ and $Q_{i+1} \leq m \leq Q_1$, we say the m-Stage Recall Complete Condition is achieved for $\mathcal{M}_i$. To this end, using $Recall@m@k$ as a guide for optimization is a relaxed version of NDCG and OPA without compromising effectiveness.

Back to the question of optimizing the pre-ranking stage in Figure 1, when the complexity of the training data $D_{train}$ produced by $\mathcal{M}_{j;2<j \leq T}$ is too high for the model capabilities of $\mathcal{M}_2$, it's considered to optimize $Recall@m@k$ is more suitable than $OPA$, $NDCG$ and $NDCG@k$. $NDCG@k$ is an relaxed metric compared to $NDCG$ that pays more attention to the correctness of the header order, in which $D_{i,j}$ and $D_{i,j}^*$ of $NDCG$ in Eq 3 are redefined as:

$$D_{i,j} = \begin{cases} p_{i,j} & \text{if } p_{i,j} \leq k \\ \infty & \text{if } p_{i,j} > k \end{cases} \qquad D_{i,j}^* = \begin{cases} p_{i,j}^* & \text{if } p_{i,j}^* \leq k \\ \infty & \text{if } p_{i,j}^* > k \end{cases} \tag{5}$$

### 4.2 Learning the Relaxed Targets via Differentiable Ranking

Many rank-based metrics are non-differentiable, which poses challenges for optimizing these metrics within the context of deep learning frameworks. Previous research on LambdaRank[3, 43] introduced a unified framework for approximate optimization of ranking metrics, which optimize the model by adopting the change

in rank metrics caused by swapping the pair as the gradient for each pair.

Next, let's first review how the pairwise loss and LambdaLoss framework can approximately optimize the model towards $OAP$, $NDCG$, $NDCG@K$ and $Recall@m@k$ respectively. To simplify the statements, let $L^{\lambda}_{(\cdot)}$ denotes the surrogate loss function for the optimization goal $(\cdot)$ on $\mathcal{M}_2$ and $\mathcal{D}[i]$ in LambdaLoss framework. For any rank metric, it can be optimized through Eq 6 under the LambdaLoss framework[3, 43].

$$L^{\lambda}_{\mathcal{R}} = \sum_{j}^{n} \sum_{h}^{n} \frac{\Delta|\mathcal{R}(j,h)|\mathbf{1}(v_i^j > v_i^h)log_2(1 + e^{-\sigma(\mathcal{M}_2^{(i,j)} - \mathcal{M}_2^{(i,h)})})}{n(n-1)/2} \tag{6}$$

where $\sigma$ is a hyper-parameter, $\mathcal{R}$ represents a rank metric, $\Delta R(j,h)$ means the $\Delta$ of $\mathcal{R}$ after swapping the position of $j$ and $h$ produced by the model, $\mathbf{1}(\cdot)$ is the indicator function. Specially, if the $\mathcal{R}$ matches the form in Eq 7, the $\Delta R(j,h)$ can be formulated as $|G_{i,j} - G_{i,h}||\frac{1}{D_{i,j}} - \frac{1}{D_{i,h}}|$, which is easy to implement under mainstream deep learning frameworks.

$$\mathcal{R}(\mathcal{D}[i], \mathcal{M}_2) = \sum_{j}^{n} \frac{G_{i,j}}{D_{i,j}} \tag{7}$$

For $L^{\lambda}_{OAP}$, $\Delta\mathcal{R}(i,h) \equiv 1$. For $L^{\lambda}_{NDCG}$ and $L^{\lambda}_{NDCG@k}$, the $G$ and $D$ is the vanilla version in $NDCG$ and $NDCG@k$. For $L^{\lambda}_{Recall@m@k}$, we can define the $G_{i,j}$ and $D_{i,j}$ as $\mathbf{1}(a_i^j) \in GS_i^k$ and $\mathbf{1}(a_i^j) \in RS_i^m$ respectively.

Although the LambdaLoss framework can provide a surrogate loss for optimizing $Recall@m@k$, it can only ensure that the gradient direction of each pair $(a_i^j, a_i^h)$ will not let $Recall@m@k$ change in a worse direction. But the size of the gradient on pair $(a_i^j, a_i^h)$ is given by heuristic info $\Delta\mathcal{R}(j,h)$. There is a concern that the overall direction of gradient optimization on the entire impression may not be very suitable for the target metric, even though the study of Wang et al.(2018)[43] showed that it can optimize a rough upper bound of the metric. For example, optimizing $L^{\lambda}_{Recall@m@k}$ may not lead to the best $Recall@m@k$, but $L^{\lambda}_{Recall@m'@k'}$ or $L^{\lambda}_{NDCG@k'}$ does, where $m \neq m'$ and $k \neq k'$. Another fact is that it is inefficient and uneconomical to perform a grid search on variants of the Lambda loss framework in order to better optimize $Recall@m@k$. And there is a lack of heuristic information to guide us in pruning grid search.

To address this challenge, we aim to create a differentiable, approximate surrogate representation of $Recall@m@k$ with a controllable approximate error bound. We intend to optimize the model end-to-end using this surrogate representation as loss directly. By doing so, the gradient direction of the surrogate loss would align with the optimization of the $Recall@m@k$ under an appropriate degree of relaxation. In other words, optimizing such a surrogate loss should yield more results in line with the optimization objective.

To design such surrogate loss, let us first introduce a description method based on permutation and matrix multiplication, which can express the sorting process and results. One sorting process $\mathcal{F}^{\uparrow}$ corresponds to a certain permutation operation, which can be

represented by a permutation matrix. For example, let $\mathcal{P}$ denote a permutation matrix, let $x = [2, 1, 4, 3]^T$ and $y = [4, 3, 2, 1]^T$ denote the origin and sorted vector, there exists a unique $\mathcal{P}$ that can represent the hard sorting for $x$. The $\mathcal{P}$ for $x$ and $y$ is shown in Eq 8; the $\mathcal{P}$, $x$ and $y$ satisfies $y = \mathcal{P}x$.

$$y = \mathcal{P}x = \begin{bmatrix} 0 & 0 & 1 & 0 \\ 0 & 0 & 0 & 1 \\ 1 & 0 & 0 & 0 \\ 0 & 1 & 0 & 0 \end{bmatrix} \begin{bmatrix} 2 \\ 1 \\ 4 \\ 3 \end{bmatrix} = \begin{bmatrix} 4 \\ 3 \\ 2 \\ 1 \end{bmatrix} \tag{8}$$

Let $\mathcal{P}^{\downarrow}_{(\cdot)}$ and $\mathcal{P}^{\uparrow}_{(\cdot)}$ denote the permutation matrices for sorting $(\cdot)$ in descending and ascending order respectively. We can also calculate rank metrics based on $\mathcal{P}$. For example, $Recall@m@k$ can be represented based on $\mathcal{P}$ as shown in Eq 9, where $\sum^{col}$ is the column sum for a 2-D matrix, $[: m]$ denotes the slice operation for a matrix which returns the first $m$ rows of the matrix, $\circ$ refers to the element-wise matrix multiplication operation.

$$Recall_{\mathcal{M}_2, \mathcal{D}[i]}@m@k = \frac{1}{k} \sum_{i}^{n} (\sum^{col} \mathcal{P}^{\downarrow}_{\mathcal{M}_2^{(i,\cdot)}}[: m] \circ \sum^{col} \mathcal{P}^{\downarrow}_{\mathbf{v}_i}[: k]) \tag{9}$$

The process to obtain a $\mathcal{P}$ for a hard sort is non-differentiable so we can't optimize Equation 9 directly under the deep learning framework. In other words, if we can give a differentiable approximation of $\mathcal{P}$, denoted as $\hat{\mathcal{P}}$, we can optimize rank metrics that are represented by $\mathcal{P}$ via $\hat{\mathcal{P}}$ directly under the deep learning framework. We notice that previous research[18, 29, 30] on differentiable sorting can produce a relaxed and differentiable permutation matrix, which can represent the hard sort approximately with a guaranteed theoretical bound. These methods often produce an unimodal row stochastic matrix or doubly-stochastic matrix. An unimodal row stochastic matrix is a square matrix in which every entry falls within the range of $[0, 1]$. It adheres to the row-stochastic property, meaning that the sum of entries in each row equals 1. Moreover, a distinguishing feature of an unimodal matrix is that within each row, there exists a unique column index associated with the maximum entry. Furthermore, a doubly-stochastic matrix extends the requirements of an unimodal row-stochastic matrix by ensuring that the sum of elements in each column equals 1. In this work, we adopt NeuralSort[18] to obtain an unimodal row stochastic matrix, which utilizes a clever mathematical transformation to ensure that $\mathcal{P}$ satisfies this property. The $i$-th row of $\hat{\mathcal{P}}$ produced by NeuralSort can be formulated as:

$$\hat{\mathcal{P}}_{\mathbf{y}}[i, :](\tau) = softmax[((L + 1 - 2i)\mathbf{y} - A_{\mathbf{y}}\mathcal{I})/\tau] \tag{10}$$

where $\mathcal{I}$ is the vector with all components equal to 1, $A_{\mathbf{y}}$ denote the matrix of absolute pairwise differences of the elements of $\mathbf{y}$ such that $A_{\mathbf{y}}[i, j] = |y_i - y_j|$, $\tau$ is the temperature of softmax which controls the approximate error of $\hat{\mathcal{P}}$ and the gradient magnitude of the inputs. $\hat{\mathcal{P}}$ will tend to $\mathcal{P}$ when $\tau$ tends to 0. A small $\tau$ leads to a small approximate error but it may cause gradient explosion, so there is a trade-off when choosing the proper $\tau$.

Utilizing $\hat{\mathcal{P}}$, we can formulate the surrogate loss function denoted as $L_{Relax}$, as illustrated in Eq 11. Our goal is to enhance the probability of $a_i^j \in RS_i^m$ for each $a_i^j \in GS_i^k$ in $\hat{\mathcal{P}}$. To achieve this, we

employ a cross-entropy-like loss function. We opt not to take Eq 9 with a replacement from $\mathcal{P}$ to $\hat{\mathcal{P}}$ as the loss because we considered that cross-entropy may be more suitable for optimizing softmax outputs, from the perspective of gradient optimization. Since the permutation matrix is only row stochastic (i.e., each row sums to 1), we incorporate the scalar $\frac{1}{m}$ into Eq.11 to ensure that the sum of values in $\hat{\mathcal{P}}$ over the selected columns, denoted by $\sum^{col} \hat{\mathcal{P}}_{\mathcal{M}_2^{(i,\cdot)}}[: m]$, does not exceed 1.

$$L_{Relax} = -\sum_j^n \{ [\sum^{col} \hat{\mathcal{P}}_{\mathbf{v}_i}[: k]] \circ [\frac{1}{m}\log(\sum^{col} \hat{\mathcal{P}}_{\mathcal{M}_2^{(i,\cdot)}}[: m])] \}_j \tag{11}$$

## 4.3 Harnessing the Relaxed Targets with Full Information Adaptively

In the preceding section, we discussed the scenario where training data is too complex, and introduced a novel surrogate loss, denoted as $L_{Relax}$, designed to optimize $Recall@m@k$ directly. However, there are cases when the training data may be less complex, and in such situations, harnessing the information from all pairs may be advantageous. Yet, determining the appropriate degree of relaxation is a challenging task, and its ideal degree may not be apparent until evaluation. Our goal is to establish a training paradigm capable of automatically and adaptively determining the appropriate degree of relaxation.

An intuition is that the optimal optimization direction of the gradient lies between the direction for optimizing toward the relaxed condition and the oracle condition. So we propose two losses that correspond to the relaxed condition and the oracle condition respectively, and leverage a multi-task learning framework to dynamically identify superior optimization directions compared to those attainable by optimizing individual losses in isolation.

Concretely, we introduce a global loss named $L_{Global}$ shown in Eq 12 which is also based on $\hat{\mathcal{P}}$. $L_{Global}$ optimizes the $OPA$ by enforcing consistency between the permutation matrix of the label and the predicted result. Specifically, it requires that the cross entropy of each row of $\hat{\mathcal{P}}_{\mathbf{v}_i}$ and $\hat{\mathcal{P}}_{\mathcal{M}_2^{(i,\cdot)}}$ tends to 1. Drawing inspiration from the uncertainty-weight method[24], we devise a comprehensive loss, as shown in Eq 13. Different from the uncertainty-weight method, we only adopt the tunable scalars for $L_{Global}$, since we hope $L_{Relax}$ serves as the primary loss and its magnitude is stable.

$$\begin{aligned} L_{Gobal} &= -\sum_j^n CE(\hat{\mathcal{P}}_{\mathbf{v}_i}[j,:], \hat{\mathcal{P}}_{\mathcal{M}_2^{(i,\cdot)}}[j,:]) \\ &= -\sum_j^n \sum_h^n [\hat{\mathcal{P}}_{\mathbf{v}_i}[j,:] \circ \log(\hat{\mathcal{P}}_{\mathcal{M}_2^{(i,\cdot)}}[j,:])]_h \end{aligned} \tag{12}$$

$$L_{total} = L_{Relax} + \frac{1}{2\alpha^2} L_{Global} + \log(|\alpha|) \tag{13}$$

This approach allows us to adaptively balance the influence of relaxation and the oracle condition during training, which is more robust to various cascade ranking scenarios. We named this approach as Adaptive Neural Ranking Framework, abbreviated as ARF. In this section, we give a simple practice of ARF that adopts

NeuralSort for relaxing the permutation matrix and utilizes a variant of the uncertainty weight method to balance the optimization of relaxed and full targets, which are basic and classic methods in differentiable sorting and multi-task learning areas. In the following section, we conduct experiments on this specific form of ARF. The flexibility of the ARF framework allows it to enjoy the benefits of differentiable sorting methods and multi-task learning methods upgrades.

## 5 EXPERIMENTS

### 5.1 Experiment Setup

To verify the effectiveness of *ARF*, we conduct both offline and online experiments.

For offline experiments, we collect two datasets from a real-world online cascade ranking system like figure 1. Both of them are collected by hierarchical random sampling in the pre-ranking space with different sampling densities, which are considered to have different learning difficulties. We record the one with higher sampling density as $industry_{hard}$, and the other with lower sampling density as $industry_{easy}$. The $v$ of $industry_{hard}$ and $industry_{easy}$ is produced by the ranking stage. The number of sampled materials per impression in $industry_{hard}$ and $industry_{easy}$ are 20 and 10 respectively. $m$ are 14 and 6, $k$ are 4 and 2 for $industry_{hard}$ and $industry_{easy}$ respectively. Note that the industrial benchmarks are created to test our method in the pre-ranking stage, so the $m$ and $k$ are corresponding to $Q_3$ and $Q_4$ in Figure 1, respectively. In addition, we also introduce two public datasets, MSLR-WEB30K[32] and Istella[14], to study the generality of our method and do some in-depth analysis. In order to train and test like in real cascade ranking scenarios, we performed some simple strategies such as truncation for pre-processing on the public datasets. Table 1 presents the statistics of these benchmarks. The details of the pre-processing and the creation process of the industrial datasets are shown in Appendix A.2 and A.1.

The architecture of the model is a basic feedforward neural network with hidden layers [1024, 512, 256]. The hidden layers adopt RELU[17] as the activation function. For all input features, we apply a log1p transformation as in [34]. The architecture is tuned based on ApproxNDCG and then fixed for all other methods. The learning rates for various benchmarks vary within the range of {1e-2, 1e-3, 1e-4}. We only tune the $\tau$ in Eq 10 and the learning rate for ARF. We conduct a grid search for $\tau$ varies from 0.1 to 10. We implement the baselines based on TF-Ranking[28]. The training process will stop when the early stop condition is reached or until 6 epochs have been trained. Note that the offline experiment settings on the industrial benchmarks are the same as the online experiments, and details are shown in appendix A.3.

To evaluate offline experiments, we adopt $Recall@m@k$ as the main metric, which is the more important metric than the ranking metrics such as $NDCG@k$ and $NDCG$ for cascade ranking systems. However, we also provide results on $NDCG@k$ and $NDCG$ in the experimental results.

For online experiments, we deploy the ARF in the pre-ranking stage of an online advertising system to study the influence of ARF in real-world cascade ranking applications. Experiment details are in section 5.5.

**Table 1: Dataset statistics.**

| Dataset | Train & Test Size | the range of materials per impression | the range of labels |
|---|---|---|---|
| $Industry_{easy}$ | 3B & 600M | 10 | [1,10] |
| $Industry_{hard}$ | 6B & 1.2B | 20 | [1,20] |
| MSLR-WEB30K | 8M & 2M | [40, 200] | [1,5] |
| Istella | 5.6M & 1.4M | [40, 200] | [1,5] |

## 5.2 Competing Methods

We compare our method with the following state-of-the-art methods in previous studies.

- **Point-wise Softmax.** According to [37], we adopt the n-class classification method by softmax as the most basic baseline. It is denoted as "Softmax" in the following.
- **RankNet.** Burges et al.(2005b)[5] propose the RankNet that constructs a classic pair-wise loss, which aims to optimize $OPA$.
- **Lambda Framework.** Based on the RankNet, Burges(2010)[3] propose the LambdaRank which can better optimize $NDCG$. Wang et al.(2018)[43] extend LambdaRank to a probabilistic framework for ranking metric optimization, which allows to define metric-driven loss functions; they also propose LambdaLoss which is considered better than vanilla LambdaRank. In the following, we denote the LambdaLoss as $L^{\lambda}_{NDCG}$. According to Lambda framework[43], we adopt its variants named $L^{\lambda}_{Recall@m@k}$ to optimize the metric $Recall@m@k$.
- **Approx NDCG.** Qin et al.(2010)[33] propose ApproxNDCG that facilitates a more direct approach to optimize NDCG. The research by Bruch et al.(2019)[2] shows that ApproxNDCG is still a strong baseline in the deep learning era.
- **LambdaLoss@K.** Jagerman et al.(2022)[22] pointed out that $NDCG@k$ cannot be optimized well based on LambdaLoss[43], and proposed a more advanced loss for $NDCG@k$. It is denoted as $L^{\lambda}_{NDCG@k}$ in this work.
- **NeuralSort.** Grover et al.(2019)[18] propose the NeuralSort that can obtain a relaxed permutation matrix for a certain sorting. Since constructing a cross-entropy loss based on $\hat{\mathcal{P}}$ is straightforward, we treat $L_{Global}$ as a baseline named "NeuralSort".

Our work focuses on the optimization of deep neural networks, so some tree model-based methods such as LambdaMART[3] are not compared. Besides, the scope of our work does not include proposing a better differentiable sorting method, so some more advanced differentiable sorting methods such as PiRank[37] and DiffSort[29, 30] are not within the scope of our comparison.

## 5.3 Main Results

Table 2 and Table 3 show the main experimental results on the public and industrial benchmarks, respectively. For industrial benchmarks, we set the $m$ and $k$ of $Recall@m@k$ according to the number of samples belonging to $Q_3$ and $Q_4$ in the benchmark. For the public benchmarks, since these data have no background information about cascade ranking, we ensure that each query has at least 15 positive documents by the pre-processing (cf., Appendix A.2) and directly specify $m = 30$ and k=15.

We can see that $L_{Relax}$ surpasses all the baselines on $Recall@m@k$ on the four benchmarks, which infers $L_{Relax}$ is an efficient surrogate loss for optimizing $Recall@m@k$. Another observation is that

**Table 2: Offline Experimental Results on Public Learning-to-rank Benchmarks.** $m$ and $k$ are 30 and 15 respectively. ∗ indicates the best results. The number in bold means that our method outperforms all the baselines on the corresponding metric. ▲ indicates the best results of the baselines.

| Method/Metric | MSLR- WEB30K | | | Istella | | |
|---|---|---|---|---|---|---|
| | Recall@m@k ↑ | NDCG@k ↑ | NDCG ↑ | Recall@m@k ↑ | NDCG@k ↑ | NDCG ↑ |
| Softmax | 0.413 | 0.416 | 0.721 | 0.357 | 0.069 | 0.359 |
| RankNet | 0.405 | 0.447 | 0.737 | 0.608 | 0.530 | 0.694 |
| ApproxNDCG | 0.440▲ | 0.504▲ | 0.765▲ | 0.628▲ | 0.588▲ | 0.736▲ |
| NeuralSort | 0.423 | 0.486 | 0.756 | 0.573 | 0.519 | 0.684 |
| $L^{\lambda}_{NDCG}$ | 0.409 | 0.453 | 0.743 | 0.626 | 0.537 | 0.703 |
| $L^{\lambda}_{NDCG@k}$ | 0.411 | 0.461 | 0.745 | 0.609 | 0.540 | 0.705 |
| $L^{\lambda}_{Recall@m@k}$ | 0.416 | 0.461 | 0.744 | 0.593 | 0.522 | 0.686 |
| $L_{Relax}$ (ours) | **0.445** | **0.511** | 0.765 | **0.644** | 0.583 | 0.729 |
| ARF (ours) | **0.446**∗ | **0.513**∗ | **0.767**∗ | **0.651**∗ | **0.598**∗ | **0.739**∗ |

the improvements in $Recall@m@k$ achieved by $L_{Relax}$ are relatively modest on the $industry_{easy}$ dataset, indicating that when the learning task is inherently simpler, the sole emphasis on relaxed targets may be less crucial. ARF, which harnesses the relaxed targets with the full targets by multi-task learning, brings further improvement and outperforms all the baselines on all evaluation metrics, which shows the effectiveness and generalization of our approach.

Among the baselines, ApproxNDCG and $L^{\lambda}_{NDCG@k}$ achieve the highest $NDCG$ and $NDCG@k$ on different benchmarks, respectively. The $Recall@m@k$ scores achieved by different baselines on various benchmarks exhibit lower consistency, which reflects the limitations of the Lambda framework in effectively optimizing $Recall@m@k$. The performance of Softmax on the Istella dataset is very poor, which may be because the label of the Istella dataset is very unbalanced. We noticed that ApproxNDCG achieves an improvement by a large margin compared to other baselines on public datasets. This may be due to the fact that our model architecture is tuned based on ApproxNDCG and the public benchmark is not big enough to erase this bias introduced by the tuning process. In contrast, the performance of ApproxNDCG on industrial benchmarks is more moderate. On larger-scale industrial datasets, each method's performance tends to be more stable. Compared to public data sets, the metrics on industrial datasets are generally higher, and the absolute differences between different methods are also smaller. This is likely due to differences in data distribution and models of the industrial and public benchmarks, such as industrial datasets have fewer materials per impression and the label repetition rate is lower when compared to the public datasets, and the model for industrial datasets is more complex (cf., Appendix A.3).

## 5.4 In-Depth Analysis

In this part, we conduct an in-depth analysis of our model. Due to the limit of time and space, we take the Istella dataset as the testbed, unless otherwise stated.

In table 2, we simply set the $m$ and $k$ as 30 and 15 during training and testing. But the $Recall@m@k$ score under other $m$ and $k$ is curious, the same goes for the methods $L_{Relax}$ and $L^{\lambda}_{Recall@m@k}$ which take $m$ and $k$ as hyper-parameters. In other words, we would like to know if the $L_{Relax}$ is a more consistent surrogate loss than the $L^{\lambda}_{Recall@m@k}$ of the Lambda framework with respect to the

**Table 3: Offline Experimental Results on Industry Benchmarks drawn from Cascade Ranking System. For $industry_{easy}$, the $m$ is 6 and the $k$ is 2. For $industry_{hard}$, the $m$ is 14 and the $k$ is 4.**

| Method/Metric | $Industry_{easy}$ | | | $Industry_{hard}$ | | |
| --- | --- | --- | --- | --- | --- | --- |
| | Recall@m@k ↑ | NDCG@k ↑ | NDCG ↑ | Recall@m@k ↑ | NDCG@k ↑ | NDCG ↑ |
| Softmax | 0.893 | 0.904 | 0.971 | 0.854 | 0.902 | 0.972 |
| RankNet | 0.928 | 0.924 | 0.978 | 0.890 | 0.932 | 0.979 |
| ApproxNDCG | 0.935 | 0.944 | 0.984 | 0.893 | 0.948 | 0.984▲ |
| NeuralSort | 0.940▲ | 0.945 | 0.984 | 0.884 | 0.947 | 0.982 |
| $L_{NDCG}^{\lambda}$ | 0.928 | 0.938 | 0.982 | 0.882 | 0.939 | 0.980 |
| $L_{NDCG@k}^{\lambda}$ | 0.935 | 0.948▲ | 0.985▲ | 0.897 | 0.950▲ | 0.984▲ |
| $L_{Recall@m@k}^{\lambda}$ | 0.931 | 0.937 | 0.979 | 0.902▲ | 0.929 | 0.978 |
| $L_{Relax}$ (ours) | **0.942** | 0.945 | 0.981 | **0.910** | 0.943 | 0.983 |
| ARF (ours) | **0.958*** | **0.949*** | **0.985*** | **0.917*** | **0.951*** | **0.989*** |

**Table 4: Online experiment results of 10% traffic for 15 days in comparison with traditional learning-to-rank methods**

| Metric | LambdaLoss@k | ARF |
| --- | --- | --- |
| Revenue | 0.0% | +1.5% |
| Conversion | 0.0% | +2.3% |

metric $Recall@m@k$. Figure 2 displays heatmaps for both $L_{Relax}$ and $L_{Recall@m@k}^{\lambda}$ with respect to the $Recall@m@k$ metric. In the heatmap, brighter blocks on the diagonal from the upper left to the lower right indicate a higher consistency between the surrogate loss and $Recall@m@k$. Compared to $L_{Recall@m@k}^{\lambda}$ under the Lambda framework, we can see that $L_{Relax}$ not only achieves overall better results under various $m$ and $k$ but also shows better consistency with $Recall@m@k$. These results further demonstrate the advancement of $L_{Relax}$. More experimental results are shown in Appendix B.

## 5.5 Online Experiments

To the best of our knowledge, ARF is the first learning-to-rank method that is designed for cascade ranking systems, and existing public datasets can't simulate the real environment of cascade ranking systems well. Therefore, it is quite important to verify the actual effect of our method in a real large-scale cascade ranking system. We applied the ARF method to the pre-ranking stage of an online advertising system and conducted an online A/B test for 15 days. The experimental traffic of the base and exp group is both 10%. Table 4 shows the online experiment results. Since we conduct the online experiments mainly to measure the real-world influence of the improvement of Recall, we only select a typical baseline method for online experiments. Our online baseline method is $LambdaLoss@k$, which also achieves good results on the industrial benchmarks. More implementation details are in Appendix A.3.

We can see from table 4 that compared to $LambdaLoss@k$, ARF has brought about a 1.5% increase in platform advertising revenue and a 2.3% increase in the number of platform user conversions. Such an improvement is considered significant in our advertising scenario.

## 6 CONCLUSION

Learning-to-rank is widely used in cascade ranking systems but traditional works usually focus on ranking metrics such as $NDCG$,

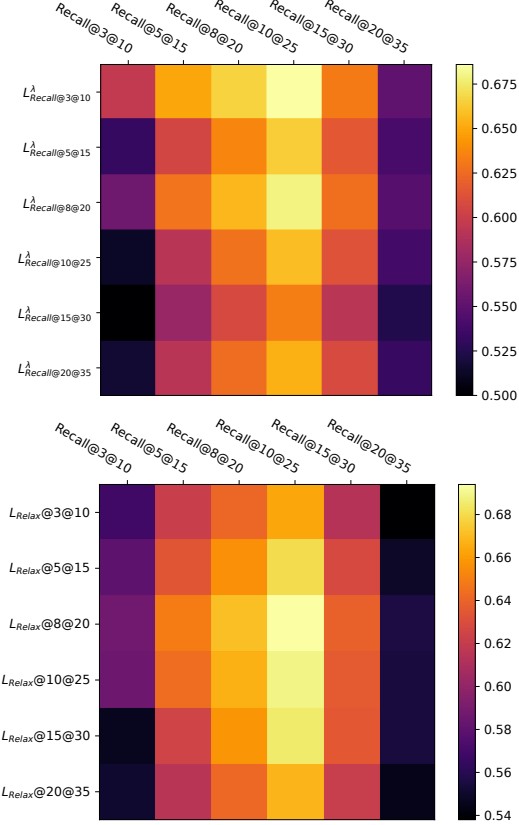

**Figure 2: The heatmap of the results on $Recall@m@k$ of $L_{Recall@m@k}^{\lambda}$ and $L_{Relax}@m@k$ under different $m$ and $k$, on Istella dataset.**

which is not an appropriate metric compared to $Recall@m@k$ for cascade ranking systems. Existing learning-to-rank methods such as LambdaFramework are not well-defined for $Recall@m@k$. Thus, we propose a novel surrogate loss named $L_{Relax}$ for $Recall@m@k$ based on differentiable sorting techniques. Considering that the combinations of data and models in different cascade ranking systems may be very diverse, we further propose the Adaptive Neural Ranking Framework that harnesses the relaxed targets $L_{Relax}$ with the information of all pairs via multi-task learning methods, to achieve robust learning-to-rank for cascade ranking.

We conduct comprehensive experiments on both public and industrial datasets, results show that our surrogate loss $L_{Relax}$ significantly outperforms the baseline on its optimization target, namely $Recall@m@k$. Profoundly, $L_{Relax}$ shows higher consistency with $Recall@m@k$ than the baselines that take $m$ and $k$ as hyper-parameters, which infers that $L_{Relax}$ is more relevant to $Recall@m@k$. ARF brings further improvements on different industrial scenarios and public datasets which infers our approach achieves a robust learning-to-rank for cascade ranking. ARF is deployed in an online advertising system, results show significant commercial value of our approach in real-world cascade ranking applications.

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

## A MORE IMPLEMENT DETAILS

### A.1 Data Creation Process of Industrial Benchmarks

To create the industrial benchmarks, we draw the data with a hierarchical random sampling strategy from the log of an online advertising system. The online advertising system adopts a four-stage cascade ranking architecture as shown in Figure 1.

For $industry_{easy}$, we draw 2 ads from $Q_4$, 4 ads from $Q_3$ and 4 ads from $Q_2$, for each impression. In order to create a more complex dataset, we increase the number of samples and the sampling ratio in spaces $Q_3$ and $Q_4$ for each impression based on $industry_{easy}$, thus creating $industry_{hard}$. Specifically, $industry_{hard}$ has 4 and 10 ads corresponding to $Q_4$ and $Q_3$ for each impression, respectively. In $industry_{hard}$, each impression has a total of 20 ads. The label of each ad is produced based on $M_3$. The ads in $Q_3$ and $Q_4$ have the predicted score of $M_3$ naturally. The ads in $Q_2$ don't have the predicted score of $M_3$ when its impression occurs normally in the online system, so we use the $M_3$ to score these ads offline after the raw training data is collected. In this way, we can give all ads a unified and fair ranking, and the ranking scores are generated by $M_3$. The rank in descending order of the ad is regarded as the training label. Based on these industrial benchmarks, we can better simulate the application effect of learning-to-rank methods in the cascade ranking system than based on the public datasets.

### A.2 Pre-processing for Public Benchmarks

Here we give more details of the pre-processing process of the public benchmarks. To evaluate $Recall@m@k$ and perform the in-depth analysis in section 5.4, we filter the query which has no more than 40 documents. And we truncate the query which has more than 200 documents. In this way, we control the number of documents per query from 40 to 200, which is similar to a stage of cascade ranking (The maximum number of documents per query is approximately 5 times the minimum value). For the truncated queries, we perform random sampling with the replacement of the documents and ensure that there are at least 15 positive documents. A positive document is a document whose score is bigger than 0.

### A.3 Online Deployment

For online experiments, the model architecture and the hyper-parameters are somewhat different from the offline experiments, which are tuned to our online environment. So we give these implementation details here.

Regarding the features, we adopt both sparse features and dense features for describing the information of the user and the ads in the online advertising system. Sparse feature means the feature whose embedding is obtained from the embedding lookup tables. Dense feature embedding is the raw values of itself. The sparse features of the user mainly include the action list of ads and user profile (e.g. age, gender and region). The action list mainly includes the action type, the frequency, the target ad, and the timestamp. The sparse features of the ads mainly include the IDs of the ad and its advertiser. The dense features of the user mainly include some embeddings produced by other pre-trained models. The dense features of the ads mainly include some side information and the multimodal features by some multimodal understanding models.

We adopt a 5-layer feed-forward neural network with units [1024, 256, 256, 256, 1]. The activation function of the hidden layer is SELU[25]. We adopt batch normalization for each hidden layer and the normalization momentum is 0.999. We employ the residual connection for each layer if its next layer has the same units. We adopt the Adam Optimizer and set the learning rate to 0.01. We perform a log1p transformation on the statistics-based dense features, following Qin et al.(2021)[34].

We train the models under an online learning paradigm. To fairly compare different methods, we cold-start train different models at the same time. We put the model online for observation after a week of training to ensure that the model has converged.

### A.4 More details for offline experiments on industrial datasets

For offline experiments, we use the same settings as the online deployment, such as the model, features, and hyperparameters.

The model architecture introduced in appendix A.3 is tuned on the RankNet method because it is our early online applicated method. The features are also just an online version drawn from the online history strategies. We do not tune the learning rate and adopt 1e-2 directly for all the methods since there is only a mild influence of the learning rate on the performance under the online learning scenario in our cascade ranking system. We only tune the $\tau$ for NeuralSort, $L_{Relax}$, and ARF, which varies from 0.1 to 10.

## B ADDITIONAL EXPERIMENTAL RESULTS

### B.1 In-Depth Analysis

Here we show additional analysis, which is not presented in the main text due to space limit (cf., Section 5.5).

Figure 3a and Figure 3b show the heatmap of $L_{Reacll@m@k}^{\lambda}$ and $L_{Relax}$ on the MSLR-WEB30K dataset. Consistently with the Istella dataset, $L_{Relax}$ yields overall better results than $L_{Recall@m@k}^{\lambda}$ under various $m$ and $k$. Compared to the Istella dataset, $L_{Recall@m@k}^{\lambda}$ and $L_{Relax}$ are not so sensitive to $m$ and $k$. This means if we want to optimize $Recall@m@k$, we do not need to fine-tune the $m$ and $k$ of the surrogate loss carefully. This is also a special case with high consistency between loss and metric. To sum all, the conclusion we can draw from Figures 3a and Figure 3b is that both $L_{Recall@m@k}^{\lambda}$ and $L_{Relax}$ show high consistency with $Recall@m@k$ on the MSLR-WEB30K data set, and the consistency between them seems to be similar (or comparable).

We also give the results of $LambdaLoss@k$ on public datasets MSLR-WEB30K and Istella, which only has $k$ as the hyper-parameter, shown in Figure 3c and Figure 3d respectively.

Compared to $LambdaLoss@k$, $L_{Relax}$ also achieves overall better results on the two public benchmarks under different $m$ and $k$, and $L_{Relax}$ also shows apparent higher consistency to $Recall@m@k$ on Istella dataset and shows comparable consistency to $Recall@m@k$ on MSLR-WEB30K dataset.

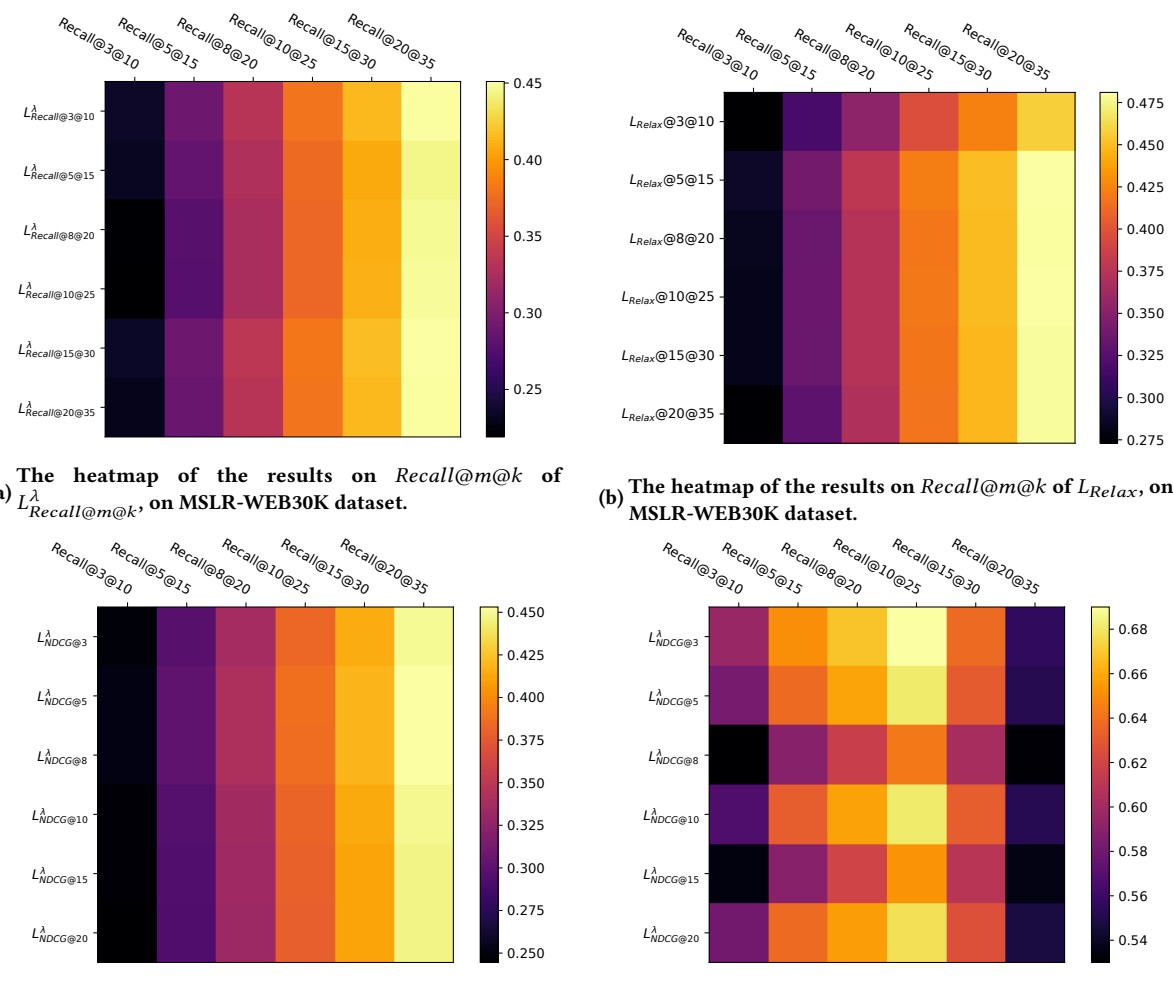

(a) The heatmap of the results on $Recall@m@k$ of $L_{Recall@m@k}^{\lambda}$, on MSLR-WEB30K dataset.

(b) The heatmap of the results on $Recall@m@k$ of $L_{Relax}$, on MSLR-WEB30K dataset.

(c) The heatmap of the results on $Recall@m@k$ of $L_{NDCG@k}^{\lambda}$, on MSLR-WEB30K dataset.

(d) The heatmap of the results on $Recall@m@k$ of $L_{NDCG@k}^{\lambda}$, on Istella dataset.

Figure 3: More heatmap results on public benchmarks.

