# OpenReview forum: "Adaptive Neural Ranking Framework: Toward Maximized Business Goal for Cascade Ranking Systems"
_ACM.org/TheWebConf/2024/Conference — TheWebConf24 Oral_

### Official Review · Reviewer_NKTY · 2023-11-14

**Novelty:** 5
**Technical Quality:** 5

**Review:**

The paper works on cascade ranking and proposes to optimize cascade ranking systems by highlighting the adaptability of optimization targets to data complexities and model capabilities. Specifically, the paper employs a multi-task learning framework to combine the optimization of relaxed and full targets and employ differentiable sorting techniques to obtain a relaxed permutation matrix and the NeuralSort method to optimize the losses jointly.

Pros:
1. The paper tries to improve the cascade ranking which is a common ranking strategy used in business systems.
2. The paper clearly discusses the disadvantages of existing works.
3. The paper is well organized and compares common baselines.

Cons:
1. The paper follows the existing idea to further optimize the model towards the relaxed condition.
2. Adopting existing techniques as the key components.

**Questions:**

1. The authors discuss two situations in cascade ranking systems, i.e., the data is simple enough and the data is complex. It is not easy to understand what are these situations and what are the main differences for cascade ranking.

**Reviewer Confidence:**

2: The reviewer is willing to defend the evaluation, but it is likely that the reviewer did not understand parts of the paper

**Scope:**

4: The work is relevant to the Web and to the track, and is of broad interest to the community

---

### Official Review · Reviewer_fmt4 · 2023-11-25

**Novelty:** 6
**Technical Quality:** 6

**Review:**

Summary: This paper introduces the Adaptive Neural Ranking Framework (ARF) as a novel approach to optimize cascade ranking systems in online advertising and recommendation scenarios. Unlike traditional learning-to-rank methods that often focus on fixed rank metrics, ARF emphasizes adaptability to varying data complexities and model capabilities. The proposed framework employs a multi-task learning approach, combining the optimization of both relaxed and full targets. To facilitate this, a permutation matrix is introduced to represent rank metrics, and differentiable sorting techniques enable the direct optimization of both relaxed and full targets within a deep learning framework. The method, named ARF, is demonstrated to be effective and generalizable through experiments on four public and industrial benchmarks. Additionally, online experiments highlight the practical application value of ARF, showcasing its significance in real-world scenarios.

Pros:
-	The introduction of the LRelaX surrogate loss is a significant contribution, offering a novel approach to optimize cascade ranking systems. LRelaX is designed to address the limitations of traditional learning-to-rank methods, providing a more effective means to optimize rankings, particularly in the context of cascade systems.

-	The proposal of ARF, utilizing a multi-task learning framework, represents a key advancement. ARF empowers the model to adaptively learn from both LRelaX and LGlobaL, introducing a comprehensive and flexible approach to tackle different scenarios in cascade ranking systems. The incorporation of an auxiliary loss function, LGlobaL, further enhances the adaptability and effectiveness of the model.

-	The paper demonstrates a thorough evaluation through comprehensive experiments conducted on four datasets, including both real-world online cascade ranking system data and standard LTR datasets. The observed superiority of LRelaX over baseline methods, especially in recall@m@k, highlights its effectiveness. Additionally, the successful deployment of the complete ARF method in an online advertising system, leading to significant improvements in business metrics, adds practical validation to the proposed approach.

-	Good quality of the write-up and well-motivated model design. By highlighting highlighting existing challenges and shortcomings, the authors establish a good motivation for the development of ARF.

Cons:

-	The introduction of the novel surrogate loss, LRelaX, by the authors is indeed innovative. However, whether ARF performs better compared to more advanced differentiable sorting methods is a discussion worth exploring. Providing insights into this aspect would be highly valuable.
-	Furthermore, to enhance the clarity and understanding of the methodology, and to emphasize key steps within the ARF multitask learning framework, it would be beneficial to include a pseudocode section or a framework diagram. This would effectively outline the algorithmic steps involved, facilitating a better comprehension of the workings of ARF for the readers.

**Questions:**

-	The authors have introduced an innovative surrogate loss, LRelax, in their work. However, investigating whether ARF outperforms more advanced differentiable sorting methods is a discussion that merits exploration. Offering insights into this aspect would contribute significantly to the paper's overall depth and impact.

-	Moreover, in order to augment the clarity and comprehension of the methodology, and to underscore crucial steps within the ARF multitask learning framework, it would be advantageous to incorporate a pseudocode section or a framework diagram. Such additions would effectively delineate the algorithmic steps, thereby facilitating a more profound understanding of the operational mechanisms of ARF for the readers.

**Reviewer Confidence:**

3: The reviewer is confident but not certain that the evaluation is correct

**Scope:**

3: The work is somewhat relevant to the Web and to the track, and is of narrow interest to a sub-community

---

### Official Review · Reviewer_X7Dn · 2023-11-27

**Novelty:** 5
**Technical Quality:** 5

**Review:**

Advantages:
The research design of this paper is rigorous, the methodology is reasonable, and it has certain inspiration and promotion for the development of the resys field.
The experimental analysis of the paper is detailed and clear, and the solution to the research question is highly convincing. The solution on the industrial dataset proves the effectiveness of the proposed ARF method.

Cons:
When comparing different methods, I am curious whether they had the same training, inference, and storage overhead, which is an important basis for evaluating algorithms. For example, whether the ARF will increase the training time of the model, and if so, how much of this additional cost.
In terms of writing, the author used an outdated template. In addition, there should be a space between the citation number and the content in the main text. Please correct these.

**Questions:**

Please see the Cons

**Reviewer Confidence:**

1: The reviewer's evaluation is an educated guess

**Scope:**

4: The work is relevant to the Web and to the track, and is of broad interest to the community

---

### Official Review · Reviewer_RXDv · 2023-11-28

**Novelty:** 4
**Technical Quality:** 4

**Review:**

Pros:
1. Tackles an important problem of optimizing business metrics in cascade ranking systems using learning-to-rank methods
2. Proposes a novel surrogate loss L_Relax for directly optimizing Recall@m@k based on differentiable sorting techniques
3. Introduces an Adaptive Neural Ranking Framework to robustly optimize both relaxed and full ranking targets using multi-task learning
4. Comprehensive experiments on public and industrial datasets demonstrate effectiveness of proposed methods

Cons:
1. Some implementation details are missing (e.g. model architecture, training methodology)
2. Additional ablation studies could lend more insight into the contribution of individual components
3. The public datasets require substantial pre-processing to conform to a cascade ranking setup. Details are given but no analysis on how alterations may impact integrity.
4. Beyond tuning temperature parameter tau, some hyperparameters seem arbitrarily set without much logic given (alpha for multi-task loss balancing). Sensitivity analysis could identify optimal configurations.
5. Online experiments use a single baseline, comparisons to other state-of-the-art methods would be informative

Overall the paper makes solid contributions in an important area. With some additional motivation, implementation details, and analysis, the work could be further strengthened.

**Questions:**

Please see above Cons section

**Reviewer Confidence:**

3: The reviewer is confident but not certain that the evaluation is correct

**Scope:**

4: The work is relevant to the Web and to the track, and is of broad interest to the community

---

### Decision · Program_Chairs · 2024-01-22

**Decision:**

Accept (Oral)

**Comment:**

The paper introduces the Adaptive Ranking Framework (ARF), a novel approach for optimizing business metrics in cascade ranking systems. This work is an important contribution to the field of learning-to-rank methods and has been demonstrated to be effective through comprehensive experiments on both public and industrial datasets. The strengths of the paper include its rigorous research design and methodology, and its detailed and clear experimental analysis. Additionally, the multi-task learning approach of ARF and its adaptability to varying data complexities and model capabilities are noteworthy advancements in the field. The practical application value of ARF is also highlighted through successful deployment in an online advertising system, resulting in significant improvements in business metrics. The paper could be further strengthened with additional details on implementation, including model architecture and training methodology. More comprehensive ablation studies to understand the contribution of individual components and sensitivity analysis on hyperparameters would also enhance the paper's depth.